# Annealed Importance Sampling with q-Paths

**Rob Brekelmans**[1][*], **Vaden Masrani**[2][*], **Thang Bui**[3],
**Frank Wood**[2], **Aram Galstyan**[1], **Greg Ver Steeg**[1], **Frank Nielsen**[4]
[1]USC Information Sciences Institute, [2]University of British Columbia, [3]UberAI, [4]Sony CSL
{brekelma,galstyan,gregv}@isi.edu, {vadmas,fwood}@cs.ubc.ca,
thang.bui@uber.com, frank.nielsen@acm.org

## Abstract

Annealed Importance Sampling (AIS) [27, 18] is the gold standard for estimating partition functions or marginal likelihoods, corresponding to importance sampling over a path of distributions between a tractable base and an unnormalized target. While AIS yields an unbiased estimator for *any* path, existing literature has been limited to the geometric mixture or moment-averaged paths associated with the exponential family and KL divergence [13]. We explore AIS using $q$-paths, which include the geometric path as a special case and are related to the homogeneous power mean, deformed exponential family, and $\alpha$-divergence [3].

## 1 Introduction

AIS [27, 18] is a method for estimating intractable normalization constants, which considers a path of intermediate distributions $\pi_t(z)$ between a tractable base distribution $\pi_0(z)$ and unnormalized target $\tilde{\pi}_T(z)$. In particular, AIS samples from a sequence of MCMC transition operators $\mathcal{T}_t(z_t|z_{t-1})$ which leave each $\pi_{\beta_t}(z) = \tilde{\pi}_{\beta_t}(z)/Z_t$ invariant to estimate the ratio $Z_T/Z_0$. As shown in Algorithm 1, we can accumulate the importance weights $w_T^{(i)} = \prod_{t=1}^{T} \tilde{\pi}_t(z_{t-1})/\tilde{\pi}_{t-1}(z_{t-1})$ along the path. Taking the expectation of $w_T^{(i)}$ over sampling chains yields an unbiased estimate of $Z_T/Z_0$ [27]. Similarly, Bidirectional Monte Carlo (BDMC) [14, 15] provides lower and upper bounds on the *log* partition function ratio $\log Z_T/Z_0$ using AIS initialized with the base or target distribution, respectively.

AIS often uses a geometric mixture path with schedule $\{\beta_t\}_{t=0}^{T}$ to anneal between $\pi_0$ and $\pi_T$,

$$\tilde{\pi}_\beta(z) = \tilde{\pi}_0(z)^{1-\beta}\,\tilde{\pi}_T(z)^\beta, \tag{1}$$

where $\pi_\beta(z) = \tilde{\pi}_\beta(z)/Z_\beta$ and $Z_\beta = \int \tilde{\pi}_0(z)^{1-\beta}\tilde{\pi}_T(z)^\beta dz$.

---

**Algorithm 1:** Annealed IS

**for** *i = 1 to N* **do**
  $z_0 \sim \pi_0(z)$
  $w^{(i)} \leftarrow Z_0$
  **for** *t = 1 to T* **do**
    $w_t^{(i)} \leftarrow w_t^{(i)} \frac{\tilde{\pi}_t(z_{t-1}^{(i)})}{\tilde{\pi}_{t-1}(z_{t-1}^{(i)})}$
    $z_t^{(i)} \sim T_t(z_t|z_{t-1}^{(i)})$
  **end**
**end**

**return** $Z_T/Z_0 \approx \frac{1}{N}\sum_N w_T^{(i)}$

---

Alternative paths have been discussed in [13, 12, 10], but may not have closed form expressions for intermediate distributions. In this work, we propose to generalize the geometric mixture path (1) using the power mean [19, 17, 11], or $q$-path,

$$\tilde{\pi}_\beta^{(q)}(z) = \left[ (1-\beta)\,\tilde{\pi}_0(z)^{1-q} + \beta\,\tilde{\pi}_T(z)^{1-q} \right]^{\frac{1}{1-q}} \tag{2}$$

As $q \to 1$, we recover the geometric mixture path as a special case. The power mean also contains as a special case the *q-logarithm* used in non-extensive thermodynamics [31, 26, 32], which allows us

---

[*]equal contribution

Deep Learning through Information Geometry Workshop (NeurIPS 2020), Vancouver, Canada.

to frame Eq. (2) in terms of the the $q$-exponential family [6]. Further, we draw connections with the $\alpha$-integration of Amari [3, 4] by showing that Eq. (2) minimizes a mixture of $\alpha$-divergences as in [3]. We describe properties of the geometric and $q$-paths in Section 2 and Section 3, respectively.

## 2  Interpretations of the Geometric Path

We give three complementary interpretations of the geometric path defined in Eq. (1), which will have generalized analogues in Section 3.

**Log Mixture**  Simply taking the logarithm of both sides of the geometric mixture (1) shows that $\tilde{\pi}_\beta$ can be obtained by taking the log-mixture of $\tilde{\pi}_0$ and $\tilde{\pi}_T$ with mixing parameter $\beta$,

$$\log \tilde{\pi}_\beta(z) = (1 - \beta) \log \tilde{\pi}_0(z) + \beta \log \tilde{\pi}_T(z) \tag{3}$$

where we may also choose to subtract a constant $\log Z_\beta$ to enforce normalization.

**Exponential Family**  Distributions along the geometric path may also be viewed as coming from an exponential family [9, 16]. In particular, we use a base measure of $\tilde{\pi}_0(z)$ and sufficient statistics $\phi(z) = \log \tilde{\pi}_T / \tilde{\pi}_0$ to rewrite Eq. (1) as

$$\pi_\beta(z) = \tilde{\pi}_0(z) \, \exp\{\beta \cdot \phi(z) - \psi(\beta)\} \tag{4}$$

where the mixing parameter $\beta$ appears as the natural parameter of the exponential family and $\psi(\beta) := \log Z_\beta$. The log-partition function or free energy $\psi(\beta)$ is convex in $\beta$ and induces [4, 29, 9] a Bregman divergence over the natural parameter space equivalent to the KL divergence $D_{KL}[\pi_{\beta'} || \pi_\beta]$.

**Variational Representation**  Grosse et al. [13] also observe that each $\pi_\beta(z)$ can be viewed as minimizing a weighted sum of KL divergences to the (normalized) base and target distributions

$$\pi_\beta(z) = \arg\min_{r(z)} (1 - \beta) \, D_{KL}[r(z) || \pi_0(z)] + \beta D_{KL}[r(z) || \pi_T(z)]. \tag{5}$$

While the optimization in Eq. (5) is over arbitrary $r(z)$, the optimal solution is the geometric mixture with mixing parameter $\beta$, which is a member of the exponential family in Eq. (4)[13, 9].

## 3  Interpretations of the $q$-Path

To anneal between $\tilde{\pi}_0$ and $\tilde{\pi}_T$, we consider the power mean with order parameter $q$ in place of the geometric average in Eq. (1). Analogously to Sec. 2 above, our generalization is associated with the deformed log mixture, $q$-exponential family, and a variational representation using the $\alpha$-divergence.

**Power Means**  Kolmogorov [19] proposed a generalized notion of the mean using any monotonic function $h(u)$, with $h(u) = u$ corresponding to the arithmetic mean and

$$\mu_h(\{w_i, u_i\}) = h^{-1}\left( \sum_i w_i \, h(u_i) \right), \tag{6}$$

where $\mu_h$ outputs a scalar given a normalized measure $\{w_i\}$ over a set of elements $\{u_i\}$ [11]. The geometric and arithmetic means are *homogeneous*, meaning they have the linear scale-free property $\mu_h(\{w_i, c \cdot u_i\}) = c \cdot \mu_h(\{w_i, u_i\})$. In order for the power mean to be homogenous, Hardy et al. [17] (pg. 68 or [3]) show that $h(u)$ must be of the form

$$h_q(u) = \begin{cases} a \cdot u^{1-q} + b & q \neq 1 \\ \log u & q = 1 \end{cases}. \tag{7}$$

which we refer to as the $q$-power mean. Notable examples of the power mean include the arithmetic mean at $q = 0$, geometric mean as $q \to 1$, and the $\min$ or $\max$ operation as $q \to \pm\infty$. For $q = \frac{1+\alpha}{2}$, $h_q(u)$ matches the $\alpha$-representation of Amari [4][5, 7].

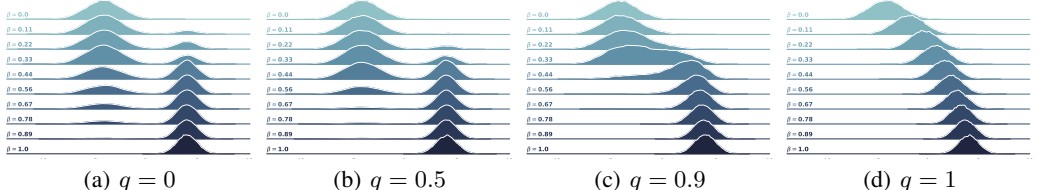

| (a) $q = 0$ | (b) $q = 0.5$ | (c) $q = 0.9$ | (d) $q = 1$ |

Figure 1: Intermediate densities between $\mathcal{N}(-4, 3)$ and $\mathcal{N}(4, 1)$ for various $q$-paths and 10 equally spaced $\beta$. The path approaches a mixture of Gaussians with weight $\beta$ at $q = 0$. For the geometric mixture ($q = 1$), intermediate $\pi_\beta$ stay within the exponential family since both $\pi_0, \pi_T$ are Gaussian.

Using the power mean to generalize geometric mean, we propose the $q$-path of intermediate unnormalized densities $\tilde{\pi}_\beta^{(q)}(z)$ for AIS. In App. A, we show that for any choice of $a$ and $b$, $h_q(u)$ yields the same power mean

$$
\tilde{\pi}_\beta^{(q)}(z) = \begin{cases} \left[(1 - \beta)\, \tilde{\pi}_0(z)^{1-q} + \beta\, \tilde{\pi}_T(z)^{1-q}\right]^{\frac{1}{1-q}} & q \neq 1 \\ \exp\left\{(1 - \beta)\, \log \tilde{\pi}_0(z) + \beta\, \log \tilde{\pi}_T(z)\right\} & q = 1 \end{cases},
\tag{8}
$$

where we have chosen $\{w_i\} = \{1 - \beta, \beta\}$ and $\{u_i\} = \{\tilde{\pi}_0, \tilde{\pi}_T\}$ in (6).

**Deformed Log Mixture**   The deformed, or $q$-logarithm [26], which plays a crucial role in non-extensive thermodynamics [31, 32], is a particular special case of $h_q(u)$ in Eq. (16), with

$$
\ln_q(u) = \frac{1}{1 - q}\left(u^{1-q} - 1\right) \qquad\qquad \exp_q(u) = \left[1 + (1 - q)\, u\right]_+^{\frac{1}{1-q}},
\tag{9}
$$

where we have also defined the $q$-exponential with $\exp_q(u) = \ln_q^{-1}(u)$ and $[x]_+ = \max\{0, x\}$ ensuring $g(u)$ is non negative. Note that $\lim_{q\to 1} \ln_q(u) = \log u$ and $\lim_{q\to 1} \exp_q(u) = \exp u$.

Applying $h_q(u) = \ln_q(u)$ to both sides of Eq. (6) or (8), we can write $\tilde{\pi}_\beta^{(q)}$ as a deformed log-mixture

$$
\ln_q \tilde{\pi}_\beta^{(q)}(z) = (1 - \beta)\, \ln_q \tilde{\pi}_0(z) + \beta\, \ln_q \tilde{\pi}_T(z)
\tag{10}
$$

with mixing weight $\beta$. We also provide detailed derivations for Eq. (10) in App. B.1.

**$q$-Exponential Family**   The $q$-exponential in Eq. (9) may be used to define a $q$-exponential family of distributions [6, 26]. Using $\theta$ as the natural parameter,

$$
\pi_\theta^{(q)}(z) = \tilde{\pi}_0(z)\, \exp_q\left\{\theta \cdot \phi_q(z) - \psi_q(\theta)\right\},
\tag{11}
$$

which recovers the standard exponential family at $q \to 1$. In App. B.2 we show that the $q$-mixture $\tilde{\pi}_\beta^{(\alpha)}$ in Eq. (8) can be rewritten in terms of the $q$-exponential family

$$
\pi_\beta^{(q)}(z) = \frac{1}{Z_\beta^{(q)}}\, \tilde{\pi}_0(z)\, \exp_q\left\{\beta \cdot \ln_q \frac{\tilde{\pi}_T(z)}{\tilde{\pi}_0(z)}\right\} \qquad\qquad Z_\beta^{(q)} = \int \tilde{\pi}_\beta^{(q)}(z)\, dz
\tag{12}
$$

with sufficient statistic $\phi_q(z) = \ln_q \tilde{\pi}_T/\tilde{\pi}_0$ and natural parameter $\beta$. The expression in (12) might be used to directly estimate the normalization constant $Z_\beta^{(q)}$ via Monte Carlo approximation.

As for the standard exponential family, the $q$-free energy $\psi_q(\theta)$ in Eq. (11) is convex in $\theta$ and can be used to construct a Bregman divergence over normalized $q$-exponential family distributions [6]. However, to normalize (12) using the $q$-free energy, a non-linear mapping $\theta(\beta)$ between parameterizations is required. This delicate issue of normalization in the $q$-exponential family has been noted in [22, 30, 26], and we provide more detailed discussion in App. B.3.

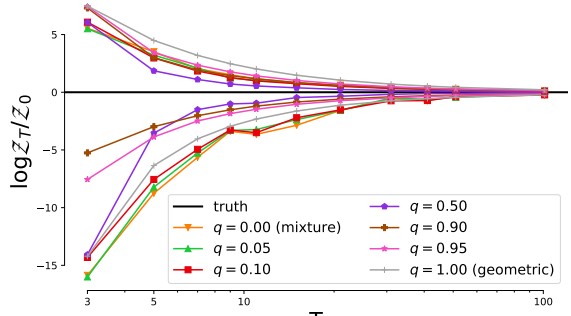

| $q$ | $Z_{\text{est}}$ ($Z_{\text{true}} = 1$) |
|---|---|
| 0.00 (mix) | $1.0136 \pm 0.0634$ |
| 0.05 | $1.0105 \pm 0.0569$ |
| 0.10 | $1.0198 \pm 0.0576$ |
| **0.90** | **$0.9975 \pm 0.0085$** |
| 0.95 | $0.9971 \pm 0.0092$ |
| 1.00 (geo) | $0.9967 \pm 0.0094$ |

Table 1: Partition Function Estimates for various $q$ and linearly spaced $T = 100$. A path with $q = 0.90$ outperforms both the mixture of Gaussians ($q = 0$) and geometric ($q = 1$) paths in terms of $Z_{\text{err}} = |Z_{\text{est}} - Z_{\text{true}}|$.

Figure 2: BDMC lower and upper bound estimates of $\log Z_T / Z_0$ by $q$-path order and number of intermediate distributions ($T$), for annealing between $\mathcal{N}(-4, 3) \to \mathcal{N}(4, 1)$.

**Variational Representation using the $\alpha$-Divergence**   Since we do not have access to normalization constants in the AIS setting, we focus on the $\alpha$-divergence [2, 4] over unnormalized measures $\tilde{q}(z)$ and $\tilde{p}(z)$. We first recall the definition,

$$D_\alpha[\tilde{q}(z) : \tilde{p}(z)] = \frac{4}{(1-\alpha^2)} \left( \frac{1-\alpha}{2} \int \tilde{q}(z)\, dz + \frac{1+\alpha}{2} \int \tilde{p}(z)\, dz - \int \tilde{q}(z)^{\frac{1-\alpha}{2}} \tilde{p}(z)^{\frac{1+\alpha}{2}}\, dz \right)$$

which is an $f$-divergence [1] for the generator $f(u) = \frac{4}{1-\alpha^2} \left( \frac{1-\alpha}{2} + \frac{1+\alpha}{2}u - u^{\frac{1+\alpha}{2}} \right)$ [5, 4]. Note that $\lim_{\alpha \to 1} D_\alpha[\tilde{q}(z) : \tilde{p}(z)] = D_{KL}[\tilde{p}(z) : \tilde{q}(z)]$ and $\lim_{\alpha \to -1} D_\alpha[\tilde{q}(z) : \tilde{p}(z)] = D_{KL}[\tilde{q}(z) : \tilde{p}(z)]$. [2]

In App. C, we follow similar derivations as Amari [3] to show that, for $q = \frac{1+\alpha}{2}$ ([4] Ch. 4), the $q$-path density $\tilde{\pi}_\beta^{(q)}$ minimizes the expected $\alpha$-divergence to the endpoints

$$\tilde{\pi}_\beta^{(q)}(z) = \arg\min_{\tilde{r}(z)} (1 - \beta) D_\alpha[\tilde{\pi}_0(z) : \tilde{r}(z)] + \beta D_\alpha[\tilde{\pi}_T(z) : \tilde{r}(z)], \tag{13}$$

where the optimization is over arbitrary $\tilde{r}(z)$. This variational representation generalizes Eq. (5), since the KL divergence is recovered (with the order of the arguments reversed) as $\alpha \to 1$ or $q \to 1$.

**Moment-Matching Procedures**   At $q = 0$, the solution to the optimization (13) corrreponds to the arithmetic mean, or mixture distribution $\tilde{\pi}_t^{(0)}(z) = (1 - \beta)\tilde{\pi}_0 + \beta\tilde{\pi}_1$. While the 'moment-averaged' AIS path [13] appears related to the $q = 0$ case, we clarify in App. C.1 that Grosse et al. [13] restrict to optimization within an exponential family of distributions. Generalizing this approach to the $\alpha$-divergence, Bui [10] follows Minka [24] (Sec. 3.1-2) to derive the moment-matching condition

$$\tilde{r}_{t,\alpha}^*(z) := \arg\min_{\tilde{r}(z)} (1 - \beta) D_\alpha[\tilde{\pi}_0(z) : \tilde{r}(z)] + \beta D_\alpha[\tilde{\pi}_T(z) : \tilde{r}(z)] \tag{14}$$

$$\implies \mathbb{E}_{\tilde{r}_*}[\phi(z)] = (1-\beta)\mathbb{E}_{\tilde{\pi}_0^\alpha r_*^{1-\alpha}}[\phi(z)] + \beta \mathbb{E}_{\tilde{\pi}_T^\alpha \tilde{r}_*^{1-\alpha}}[\phi(z)] \tag{15}$$

where $\tilde{r}(z)$ comes from an exponential family with sufficient statistics $\phi(z)$.

However, we note that our $q$-path is more general than these approaches, since the optimization in Eq. (13) is over all unnormalized distributions. Unlike the moment matching conditions above, our closed form expression for $\tilde{\pi}_\beta^{(q)}$ can be directly used as an energy function for MCMC sampling.

## 4   Experiments

We consider $q$-paths between $\pi_0 = \mathcal{N}(-4, 3)$ and $\pi_T = \mathcal{N}(4, 1)$ to estimate $Z_T / Z_0 = 1$, and use parallel runs of Hamiltonian Monte Carlo (HMC) [28] to obtain accurate, independent samples from $\tilde{\pi}_t^{(q)}(z)$ linearly spaced between $\beta_0 = 0$ and $\beta_T = 1$. For all experiments, we use 10k samples from each intermediate distribution and average results across 20 seeds.

---

[2] We extend to unnormalized measures using $D_{KL}[\tilde{q}(z) : \tilde{p}(z)] = D_{KL}[q(z) : p(z)] - \int \tilde{q}(z) dz + \int \tilde{p}(z) dz$.

In Fig. 2, we report BDMC upper and lower bound estimates of $\log Z_T/Z_1$ for various $q$ and $T$. We observe that the choice of $q$ can impact performance, with $q = 0.9$ obtaining tighter estimates at small $T$ and $q = 0.5$ converging more quickly as $T$ increases. Both outperform the baseline geometric path at $q = 1$. In Table 1, we estimate $Z_T/Z_0$ using AIS for $T = 100$, and observe that our the $q = 0.9$ path can achieve a lower error than the geometric path.

Finally, in App. E, we provide additional analysis for annealing between two Student-$t$ distributions. The Student-$t$ family can be shown to correspond to a $q$-exponential family [21], with the same sufficient statistics as a Gaussian, and a degrees of freedom parameter $\nu$ that induces heavier tails and sets the value of $q$. As $q \to 1$ or $\nu \to \infty$, the standard Gaussian is recovered. In Fig. 3-4, we compare annealing between two Student-$t$ distributions in the $q = 2$ family to the Gaussian case of $q = 1$, and observe that the same $q$-path can induce different qualitative behavior based on properties of the endpoint distributions.

## 5   Conclusion

In this work, we propose $q$-paths to generalize the geometric mixture path commonly used in AIS, and show that modifying the path can improve AIS and BDMC for a fixed mixing schedule on a toy Gaussian example. We interpreted our $q$-paths using the deformed logarithm, $q$-exponential family, and $\alpha$-divergences, which may suggest further connections in non-extensive thermodynamics and information geometry. Choosing a schedule for a given $q$-path, understanding how the choice of $q$ depends on properties of the initial and target distributions, and exploring the use of $q$-paths in related methods such as the thermodynamic variational objective (TVO) [20, 9] remain interesting directions for future work.

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

# A  Abstract Mean is Invariant to Affine Transformations

In this section, we show that $h_q(u)$ is invariant to affine transformations. That is, for any choice of $a$ and $b$,

$$h_q(u) = \begin{cases} a \cdot u^{1-q} + b & q \neq 1 \\ \log u & q = 1 \end{cases} \tag{16}$$

yields the same expression for the abstract mean $\mu_{h_\alpha}$. First, we note the expression for the inverse $h_q^{-1}(u)$ at $q \neq 1$

$$h_q^{-1}(u) = \left( \frac{u-b}{a} \right)^{\frac{1}{1-q}}. \tag{17}$$

Recalling that $\sum_i w_i = 1$, the abstract mean then becomes

$$\mu_{h_q}(\{w_i\}, \{u_i\}) = h_q^{-1} \left( \sum_i w_i h_q(u_i) \right) \tag{18}$$

$$= h_q^{-1} \left( a \left( \sum_i w_i u_i^{1-q} \right) + b \right) \tag{19}$$

$$= \left( \sum_i w_i u_i^{1-q} \right)^{\frac{1}{1-q}} \tag{20}$$

which is independent of both $a$ and $b$.

# B  Derivations of the $q$-Path

## B.1  Deformed Log Mixture

In this section, we show that the unnormalized $\ln_q$ mixture

$$\ln_q \tilde{\pi}_\beta^{(q)}(z) = (1-\beta) \ln_q \tilde{\pi}_0(z) + \beta \ln_q \tilde{\pi}_1(z) \tag{21}$$

reduces to the form of the $q$-path intermediate distribution in (2) and (8). Taking $\exp_q$ of both sides,

$$\tilde{\pi}_\beta^{(q)}(z) = \exp_q \left\{ (1-\beta) \ln_q \tilde{\pi}_0(z) + \beta \ln_q \tilde{\pi}_1(z) \right\}$$

$$= [1 + (1-q) (\ln_q \tilde{\pi}_0(z) + \beta (\ln_q \tilde{\pi}_1(z) - \ln_q \tilde{\pi}_0(z)))]_+^{\frac{1}{1-q}}$$

$$= \left[ 1 + (1-q) \frac{1}{1-q} \left( \tilde{\pi}_0(z)^{1-q} - 1 + \beta (\tilde{\pi}_1(z)^{1-q} - 1 - \tilde{\pi}_0(z)^{1-q} + 1) \right) \right]_+^{\frac{1}{1-q}}$$

$$= \left[ 1 + \tilde{\pi}_0(z)^{1-q} - 1 + \beta \left( \tilde{\pi}_1(z)^{1-q} - \tilde{\pi}_0(z)^{1-q} \right) \right]_+^{\frac{1}{1-q}}$$

$$= \left[ \tilde{\pi}_0(z)^{1-q} + \beta \tilde{\pi}_1(z)^{1-q} - \beta \tilde{\pi}_0(z)^{1-q} \right]_+^{\frac{1}{1-q}}$$

$$= \left[ (1-\beta) \tilde{\pi}_0(z)^{1-q} + \beta \tilde{\pi}_1(z)^{1-q} \right]_+^{\frac{1}{1-q}}$$

## B.2  $q$-Exponential Family

Here, we show that the unnormalized $q$-path reduces to a form of the $q$-exponential family

$$\tilde{\pi}_\beta^{(q)}(z) = \left[(1-\beta)\tilde{\pi}_0(z)^{1-q} + \beta\tilde{\pi}_1(z)^{1-q}\right]^{\frac{1}{1-q}} \tag{22}$$

$$= \left[\tilde{\pi}_0(z)^{1-q} + \beta\big(\tilde{\pi}_1(z)^{1-q} - \tilde{\pi}_0(z)^{1-q}\big)\right]^{\frac{1}{1-q}} \tag{23}$$

$$= \tilde{\pi}_0(z)\left[1 + \beta\left(\left(\frac{\tilde{\pi}_1(z)}{\tilde{\pi}_0(z)}\right)^{1-q} - 1\right)\right]^{\frac{1}{1-q}} \tag{24}$$

$$= \tilde{\pi}_0(z)\left[1 + (1-q)\,\beta\,\ln_q\left(\frac{\tilde{\pi}_1(z)}{\tilde{\pi}_0(z)}\right)\right]^{\frac{1}{1-q}} \tag{25}$$

$$= \tilde{\pi}_0(z)\exp_q\left\{\beta\cdot\ln_q\left(\frac{\tilde{\pi}_1(z)}{\tilde{\pi}_0(z)}\right)\right\}. \tag{26}$$

Defining $\phi(z) = \ln_q\frac{\tilde{\pi}_1(z)}{\tilde{\pi}_0(z)}$ and introducing a multiplicative normalization factor $Z_q(\beta)$, we arrive at

$$\pi_\beta^{(q)}(z) = \frac{1}{Z_q(\beta)}\,\tilde{\pi}_0(z)\exp_q\left\{\beta\cdot\phi(z)\right\} \qquad Z_q(\beta) := \int\tilde{\pi}_0(z)\exp_q\left\{\beta\cdot\phi(z)\right\}\,dz. \tag{27}$$

## B.3  Normalization in q-Exponential Families

The $q$-exponential family can also be written using the $q$-free energy $\psi_q(\theta)$ for normalization [6, 26],

$$\pi_\theta^{(q)}(z) = \pi_0(z)\,\exp_q\left\{\theta\cdot\phi(z) - \psi_q(\theta)\right\}. \tag{28}$$

However, since $\exp_q\{x+y\} = \exp_q\{y\}\cdot\exp_q\{\frac{x}{1+(1-q)y}\}$ (see [30] or App. D below) instead of $\exp\{x+y\} = \exp\{x\}\cdot\exp\{y\}$ for the standard exponential, we can not easily move between these ways of writing the $q$-family [22].

Mirroring the derivations of Naudts [26] pg. 108, we can rewrite (28) using the above identity for $\exp_q\{x+y\}$, as

$$\pi_\theta^{(q)}(z) = \pi_0(z)\,\exp_q\{\theta\cdot\phi(z) - \psi_q(\theta)\} \tag{29}$$

$$= \pi_0(z)\,\exp_q\{-\psi_q(\theta)\}\exp_q\left\{\frac{\theta\cdot\phi(z)}{1+(1-q)(-\psi_q(\theta))}\right\} \tag{30}$$

Our goal is to express $\pi_\theta^{(q)}(z)$ using a normalization constant $Z_\beta^{(q)}$ instead of the $q$-free energy $\psi_q(\theta)$. While the exponential family allows us to freely move between $\psi(\theta)$ and $\log Z_\theta$, we must adjust the natural parameters (from $\theta$ to $\beta$) in the $q$-exponential case. Defining

$$\beta = \frac{\theta}{1+(1-q)(-\psi_q(\theta))} \tag{31}$$

$$Z_\beta^{(q)} = \frac{1}{\exp_q\{-\psi_q(\theta)\}} \tag{32}$$

we can obtain a new parameterization of the $q$-exponential family, using parameters $\beta$ and multiplicative normalization constant $Z_\beta^{(q)}$,

$$\pi_\beta^{(q)}(z) = \frac{1}{Z_\beta^{(q)}}\pi_0(z)\,\exp_q\{\beta\cdot\phi(z)\} \tag{33}$$

$$= \pi_0(z)\,\exp_q\left\{\theta\cdot\phi(z) - \psi_q(\theta)\right\} = \pi_\theta^{(q)}(z). \tag{34}$$

See Matsuzoe et al. [22], Suyari et al. [30], and Naudts [26] for more detailed discussion of normalization in deformed exponential families.

## C  Minimizing $\alpha$-divergences

Amari [3] show that the $\alpha$-mixture $\pi_{\alpha_t}$ minimizes the expected divergence to a single point for *normalized* measures. We repeat similar derivations but for the case of unnormalized $\{\tilde{p}_i\}$ and $\tilde{r}(z)$

$$\tilde{\pi}_\alpha(z) = \arg\min_{\tilde{r}(z)} \sum_{i=1}^{N} w_i \, D_\alpha[\tilde{p}_i(z) : \tilde{r}(z)] \tag{35}$$

$$\text{where} \quad \tilde{\pi}_\alpha(z) = \Big(\sum_{i=1}^{N} w_i \, \tilde{p}_i(z)^{\frac{1-\alpha}{2}}\Big)^{\frac{2}{1-\alpha}} \tag{36}$$

*Proof.*

$$\frac{d}{d\tilde{r}} \sum_{i=1}^{N} w_i \, D_\alpha[\tilde{p}_i(z) : \tilde{r}(z)] = \frac{d}{d\tilde{r}} \frac{4}{1-\alpha^2} \sum_{i=1}^{N} w_i \Big(-\int \tilde{p}_i(z)^{\frac{1-\alpha}{2}} \, \tilde{r}(z)^{\frac{1+\alpha}{2}} dz + \frac{1+\alpha}{2} \int \tilde{r}(z)dz\Big) \tag{37}$$

$$0 = \frac{4}{1-\alpha^2}\Big(-\frac{1+\alpha}{2}\sum_{i=1}^{N} w_i \, \tilde{p}_i(z)^{\frac{1-\alpha}{2}} \, \tilde{r}(z)^{\frac{1+\alpha}{2}-1} + \frac{1+\alpha}{2}\Big) \tag{38}$$

$$-\frac{2}{1-\alpha} = -\frac{2}{1-\alpha}\sum_{i=1}^{N} w_i \, \tilde{p}_i(z)^{\frac{1-\alpha}{2}} \tilde{r}(z)^{-\frac{1-\alpha}{2}} \tag{39}$$

$$\tilde{r}(z)^{\frac{1-\alpha}{2}} = \sum_{i=1}^{N} w_i \, \tilde{p}_i(z)^{\frac{1-\alpha}{2}} \tag{40}$$

$$\tilde{r}(z) = \Big(\sum_{i=1}^{N} w_i \, \tilde{p}_i(z)^{\frac{1-\alpha}{2}}\Big)^{\frac{2}{1-\alpha}} \tag{41}$$

$\square$

This result is similar to a general result about Bregman divergences in Banerjee et al. [8] Prop. 1. although $D_\alpha$ is not a Bregman divergence over normalized distributions.

### C.1  Arithmetic Mean ($q = 0$)

The moment-averaging path from Grosse et al. [13] is *not* a special case of the $\alpha$-mean path of Amari [3]. While both minimize a convex combination of reverse KL divergences, Grosse et al. [13] minimize within the constrained space of exponential families , while Amari [3] optimizes over *all* normalized distributions.

More formally, consider minimizing the functional

$$J[r] = (1-\beta)\int \pi_0(z) \log \frac{\pi_0(z)}{r(z)}dz + \beta \int \pi_1(z) \log \frac{\pi_1(z)}{r(z)}dz \tag{42}$$

$$= \text{const} - \int [(1-\beta)\pi_0(z) + \beta\pi_1(z)] \log r(z)dz \tag{43}$$

We will show how Grosse et al. [13] and Amari [3] minimize (43).

**Solution within Exponential Family**  Grosse et al. [13] constrains $r(z) = \frac{1}{Z(\theta)}h(z)\exp(\theta^T g(z))$ to be a (minimal) exponential family model and minimizes (43) w.r.t $r$'s natural parameters $\theta$ (cf. [13] appendix 2.2):

$$\theta_i^* = \arg\min_\theta J(\theta) \tag{44}$$

$$= \arg\min_\theta \left(-\int [(1-\beta)\pi_0(z) + \beta\pi_1(z)]\left[\log h(z) + \theta^T g(z) - \log Z(\theta)\right]dz\right) \tag{45}$$

$$= \arg\min_\theta \left(\log Z(\theta) - \int [(1-\beta)\pi_0(z) + \beta\pi_1(z)]\,\theta^T g(z)dz + \text{const}\right) \tag{46}$$

where the last line follows because $\pi_0(z)$ and $\pi_1(z)$ are assumed to be correctly normalized. Then to arrive at the moment averaging path, we compute the partials $\frac{\partial J(\theta)}{\partial \theta_i}$ and set to zero:

$$\frac{\partial J(\theta)}{\partial \theta_i} = \mathbb{E}_r[g_i(z)] - (1-\beta)\,\mathbb{E}_{\pi_0}[g_i(z)] - \beta\,\mathbb{E}_{\pi_1}[g_i(z)] = 0 \tag{47}$$

$$\mathbb{E}_r[g_i(z)] = (1-\beta)\,\mathbb{E}_{\pi_0}[g_i(z)] - \beta\,\mathbb{E}_{\pi_1}[g_i(z)] \tag{48}$$

where we have used the exponential family identity $\frac{\partial \log Z(\theta)}{\partial \theta_i} = \mathbb{E}_{r_\theta}[g_i(z)]$ in the first line.

**General Solution**    Instead of optimizing in the space of minimal exponential families, Amari [3] instead adds a Lagrange multiplier to (43) and optimizes $r$ directly (cf. [3] eq. 5.1 - 5.12)

$$r^* = \arg\min_r J'[r] \tag{49}$$

$$= \arg\min_r J[r] + \lambda\left(1 - \int r(z)dz\right) \tag{50}$$

Eq. (50) can be minimized using the Euler-Lagrange equations or using the identity

$$\frac{\delta f(x)}{\delta f(x')} = \delta(x - x') \tag{51}$$

from [23]. We compute the functional derivative of $J'[r]$ using (51) and solve for $r$:

$$\frac{\delta J'[r]}{\delta r(z)} = -\int \left[(1-\beta)\pi_0(z') + \beta\pi_1(z')\right]\frac{1}{r(z')}\frac{\delta r(z')}{\delta r(z)}dz' - \lambda\int\frac{\delta r(z')}{\delta r(z)}dz' \tag{52}$$

$$= -\int \left[(1-\beta)\pi_0(z') + \beta\pi_1(z')\right]\frac{1}{r(z')}\delta(z - z')dz' - \lambda\int \delta(z - z')dz' \tag{53}$$

$$= -\left[(1-\beta)\pi_0(z) + \beta\pi_1(z)\right]\frac{1}{r(z)} - \lambda = 0 \tag{54}$$

Therefore

$$r(z) \propto \left[(1-\beta)\pi_0(z) + \beta\pi_1(z)\right], \tag{55}$$

which corresponds to our $q$-path at $q = 0$, or $\alpha = -1$ in Amari [3]. Thus, while both Amari [3] and Grosse et al. [13] start with the same objective, they arrive at different optimum because they optimize over different spaces.

## D    Sum and Product Identities for $q$-Exponentials

In this section, we prove two lemmas which are useful for manipulation expressions involving $q$-exponentials, for example in moving between Eq. (29) and Eq. (30) in either direction.

**Lemma 1.** *Sum identity*

$$\exp_q\left(\sum_{n=1}^N x_n\right) = \prod_{n=1}^N \exp_q\left(\frac{x_n}{1 + (1-q)\sum_{i=1}^{n-1} x_i}\right) \tag{56}$$

**Lemma 2.** *Product identity*

$$\prod_{n=1}^N \exp_q(x_n) = \exp_q\left(\sum_{n=1}^N x_n \cdot \prod_{i=1}^{n-1}(1 + (1-q)x_i)\right) \tag{57}$$

## D.1 Proof of Lemma 1

*Proof.* We prove by induction. The base case ($N = 1$) is satisfied using the convention $\sum_{i=a}^{b} x_i = 0$ if $b < a$ so that the denominator on the RHS of Eq. (56) is 1. Assuming Eq. (56) holds for $N$,

$$\exp_q \left( \sum_{n=1}^{N+1} x_n \right) = \left[ 1 + (1-q) \sum_{n=1}^{N+1} x_n \right]_+^{1/(1-q)} \tag{58}$$

$$= \left[ 1 + (1-q) \left( \sum_{n=1}^{N} x_n \right) + (1-q)x_{N+1} \right]_+^{1/(1-q)} \tag{59}$$

$$= \left[ \left( 1 + (1-q) \sum_{n=1}^{N} x_n \right) \left( 1 + (1-q)\frac{x_{N+1}}{1 + (1-q) \sum_{n=1}^{N} x_n} \right) \right]_+^{1/(1-q)} \tag{60}$$

$$= \exp_q \left( \sum_{n=1}^{N} x_n \right) \exp_q \left( \frac{x_{N+1}}{1 + (1-q) \sum_{n=1}^{N} x_n} \right) \tag{61}$$

$$= \prod_{n=1}^{N+1} \exp_q \left( \frac{x_n}{1 + (1-q) \sum_{i=1}^{n-1} x_i} \right) \text{ (using the inductive hypothesis)} \tag{62}$$

$\square$

## D.2 Proof of Lemma 2

*Proof.* We prove by induction. The base case ($N = 1$) is satisfied using the convention $\prod_{i=a}^{b} x_i = 1$ if $b < a$. Assuming Eq. (57) holds for $N$, we will show the $N + 1$ case. To simplify notation we define $y_N := \sum_{n=1}^{N} x_n \cdot \prod_{i=1}^{n-1} (1+ = (1-q)x_i)$. Then,

$$\prod_{n=1}^{N+1} \exp_q(x_n) = \exp_q(x_1) \left( \prod_{n=2}^{N+1} \exp_q(x_n) \right) \tag{63}$$

$$= \exp_q(x_0) \left( \prod_{n=1}^{N} \exp_q(x_n) \right) \qquad \text{(reindex } n \to n - 1)$$

$$= \exp_q(x_0) \exp_q(y_N) \qquad \text{(inductive hypothesis)}$$

$$= \left[ (1 + (1-q) \cdot x_0)(1 + (1-q) \cdot y_N) \right]_+^{1/(1-q)} \tag{64}$$

$$= \left[ 1 + (1-q) \cdot x_0 + (1 + (1-q) \cdot x_0)(1-q) \cdot y_N \right]_+^{1/(1-q)} \tag{65}$$

$$= \left[ 1 + (1-q) \left( x_0 + (1 + (1-q) \cdot x_0)y_N \right) \right]_+^{1/(1-q)} \tag{66}$$

$$= \exp_q \left( x_0 + (1 + (1-q) \cdot x_0)y_N \right) \tag{67}$$

Next we use the definition of $y_N$ and rearrange

$$= \exp_q \left( x_0 + (1 + (1-q) \cdot x_0) \left( x_1 + x_2(1 + (1-q) \cdot x_1) + ... + x_N \cdot \prod_{i=1}^{N-1} (1 + (1-q) \cdot x_i) \right) \right)$$

$$= \exp_q \left( \sum_{n=0}^{N} x_n \cdot \prod_{i=1}^{n-1} (1 + (1-q)x_i) \right). \tag{68}$$

Then reindexing $n \to n + 1$ establishes

$$\prod_{n=1}^{N+1} \exp_q(x_n) = \exp_q \left( \sum_{n=1}^{N+1} x_n \cdot \prod_{i=1}^{n-1} (1 + (1-q)x_i) \right). \tag{69}$$

$\square$

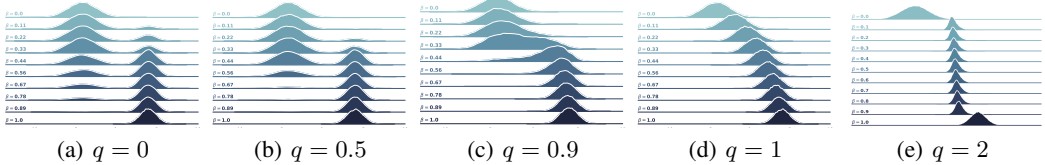

(a) $q = 0$      (b) $q = 0.5$      (c) $q = 0.9$      (d) $q = 1$      (e) $q = 2$

Figure 3: Intermediate densities between $\mathcal{N}(-4, 3)$ and $\mathcal{N}(4, 1)$ for various $q$-paths and 10 equally spaced $\beta$. The path approaches a mixture of Gaussians with weight $\beta$ at $q = 0$. For the geometric mixture ($q = 1$), intermediate $\pi_\beta$ stay within the exponential family since both $\pi_0$, $\pi_T$ are Gaussian.

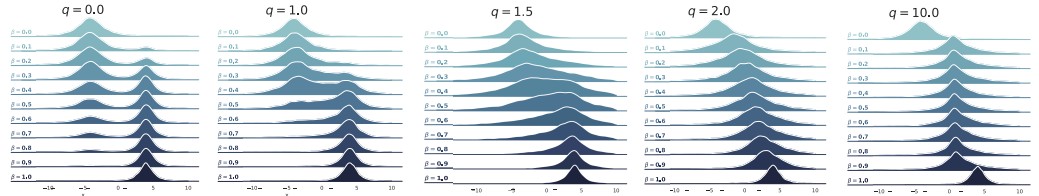

Figure 4: Intermediate densities between Student-$t$ distributions, $t_{\nu=1}(-4, 3)$ and $t_{\nu=1}(4, 1)$ for various $q$-paths and 10 equally spaced $\beta$, Note that $\nu = 1$ corresponds to $q = 2$, so that the $q = 2$ path stays within the $q$-exponential family.

## E    Annealing between Student-$t$ Distributions

### E.1    Student-$t$ Distributions and $q$-Exponential Family

The Student-$t$ distribution appears in hypothesis testing with finite samples, under the assumption that the sample mean follows a Gaussian distribution. In particular, the degrees of freedom parameter $\nu = n-1$ can be shown to correspond to an order of the $q$-exponential family with $\nu = (3-q)/(q-1)$ (in 1-d), so that the choice of $q$ is linked to the amount of data observed.

We can first write the multivariate Student-$t$ density, specified by a mean vector $\mu$, covariance $\Sigma$, and degrees of freedom parameter $\nu$, in $d$ dimensions, as

$$t_\nu(x|\mu, \Sigma) = \frac{1}{Z(\nu, \Sigma)} \left[1 + \frac{1}{\nu}(x - \mu)^T \Sigma^{-1}(x - \mu)\right]^{-\left(\frac{\nu+d}{2}\right)} \tag{70}$$

where $Z(\nu, \Sigma) = \Gamma(\frac{\nu+d}{2})/\Gamma(\frac{\nu}{2}) \cdot |\Sigma|^{-1/2} \nu^{-\frac{d}{2}} \pi^{-\frac{d}{2}}$. Note that $\nu > 0$, so that we only have positive values raised to the $-(\nu + d)/2$ power, and the density is defined on the real line.

The power function in (70) is already reminiscent of the $q$-exponential, while we have first and second moment sufficient statistics as in the Gaussian case. We can solve for the exponent, or order parameter $q$, that corresponds to $-(\nu + d)/2$ using $-\left(\frac{\nu+d}{2}\right) = \frac{1}{1-q}$. This results in the relations

$$\nu = \frac{d - dq + 2}{q - 1} \qquad \text{or} \qquad q = \frac{\nu + d + 2}{\nu + d} \tag{71}$$

We can also rewrite the $\nu^{-1}(x - \mu)^T \Sigma^{-1}(x - \mu)$ using natural parameters corresponding to $\{x, x^2\}$ sufficient statistics as in the Gaussian case (see, e.g. Matsuzoe and Wada [21] Example 4).

Note that the Student-$t$ distribution has heavier tails than a standard Gaussian, and reduces to a multivariate Gaussian as $q \to 1$ and $\exp_q(u) \to \exp(u)$. This corresponds to observing $n \to \infty$ samples, so that the sample mean and variance approach the ground truth [25].

### E.2    Annealing between 1-d Student-$t$ Distributions

Since the Student-$t$ family generalizes the Gaussian distribution to $q \neq 1$, we can run a similar experiment annealing between two Student-$t$ distributions. We set $q = 2$, which corresponds to $\nu = 1$ with $\nu = (3 - q)/(q - 1)$, and use the same mean and variance as the Gaussian example in Fig. 4, with $\pi_0(z) = t_{\nu=1}(-4, 3)$ and $\pi_1(z) = t_{\nu=1}(4, 1)$.

We visualize the results in Fig. 4. For this special case of both endpoint distributions within a parametric family, we can ensure that the $q = 2$ path stays within the $q$-exponential family of Student-$t$ distributions. We make a similar observation for the Gaussian case and $q = 1$ in Fig. 3. Comparing the $q = 0.5$ and $q = 0.9$ Gaussian path with the $q = 1.0$ and $q = 1.5$ path, we observe that mixing behavior appears to depend on the relation between the $q$-path parameter and the order of the $q$-exponential family of the endpoints.

As $q \to \infty$, the power mean (6) approaches the $\min$ operation as $1 - q \to -\infty$. In the Gaussian case, we see that, even at $q = 2$, intermediate densities for all $\beta$ appear to concentrate in regions of low density under both $\pi_0$ and $\pi_T$. However, for the heavier-tailed Student-$t$ distributions, we must raise the $q$-path parameter significantly to observe similar behavior.

