# OpenReview forum: "Annealed Importance Sampling with q-Paths"
_NeurIPS.cc/2020/Workshop/DL-IG — NeurIPSW 2020: DL-IG Oral_

### Official Review · AnonReviewer2 · 2020-10-30
**Nice result**

**Rating:** 9
**Confidence:** 5

**Review:**

The submission revisits the path used for Annealed Importance Sampling and considers paths that interpolate between the two distributions using a generalized mean.  This gives them a family of q-paths, with properties analogous to the standard geometric mixture paths but with generalized notions.  E.g. generalized exponential families over ordinary ones, or having the intermediate states be the solution to a weighted alpha-divergence minimization rather than a weighted KL divergence.

I think the paper is nice, it's a neat result and its is presented well.  It's shown that on a simple problem these broader notions of paths can show improved convergence and more accurate measurements than the typically chosen paths.

I think it's a great workshop submission and a neat result, don't think I have any complaints.

---

### Official Review · AnonReviewer1 · 2020-11-01

**Rating:** 8
**Confidence:** 4

**Review:**

This paper explores annealed importance sampling (AIS) using a power-mean of the initial and target distributions. Geometric mean of the initial and target is typically used to estimate the log-partition function via AIS, using a power-mean has the benefit that the interpolated distribution can belong to a different family, as shown by the authors in Fig. 1. The paper further discusses connections with log-mixtures, alpha-divergences and q-exponential families. Experimental results are provided that compute the ratio of the partition functions between two Gaussians with Hamiltonian Monte Carlo to obtain samples from the interpolants. The Appendix develops q-paths further.

---

### Decision · Program_Chairs · 2020-11-07

Accept (Oral)